# Impact Estimation of Unplanned Urban Rail Disruptions on Public Transport Passengers: A Multi-Agent Based Simulation Approach

**DOI:** 10.3390/ijerph19159052

**Published:** 2022-07-25

**Authors:** Chengli Cong, Xuan Li, Shiwei Yang, Quan Zhang, Lili Lu, Yang Shi

**Affiliations:** 1School of Maritime and Transportation, Ningbo University, Ningbo 315211, China; 2011084001@nbu.edu.cn (C.C.); maxchanging@foxmail.com (S.Y.); 2011084018@nbu.edu.cn (Q.Z.); lulili@nbu.edu.cn (L.L.); 2Jiangsu Province Collaborative Innovation Center of Modern Urban Traffic Technologies, Southeast University, Road #2, Nanjing 211189, China; 3Ningbo Urban Planning & Design Institute, Ningbo 315042, China; shiyang_213@foxmail.com

**Keywords:** urban rail transit, unplanned disruption, impact estimation, multi-agent simulation

## Abstract

Once unplanned urban rail disruptions occur, it is essential to evaluate the impacts on public transport passengers since impact estimation results enable transit agencies to verify whether alternative transit services have adequate capacity to evacuate the affected rail passengers and to adopt effective emergency measures in response to the disruptions. This paper focuses on estimating the impacts of unplanned rail line segment disruptions on rail passengers as well as original bus passengers, as the latter are overlooked in existing studies. A method of identifying affected rail passengers based on passenger tap-in time is proposed, which is helpful for evaluating the scale and origin-destination distribution of the affected passengers. Passengers’ response behaviors are analyzed and modeled in a multi-agent simulation system. The system realizes the simulation of the multimodal evacuation process, in which a rule-based logit model is employed to describe passengers’ travel selection behavior and the Monte Carlo method is utilized to address the issue of uncertainty in passengers’ travel selection. In particular, the original bus passengers are integrated into the simulation and interact with rail passengers. Finally, some indicators assessing the impacts on rail passengers and bus passengers are presented, and a case study based on the Ningbo urban rail transit network is conducted.

## 1. Introduction

Urban rail transit (URT) shows a trend of large-scale and rapid development for cities worldwide. Due to continuous expansion, high-intensity operation and aging equipment, unplanned service disruptions occur frequently, causing negative impacts on a large number of passengers as well as the transportation system, not limited to the URT system. What exactly is the influence of rail disruption on passengers and the transportation system? It is essential for transit agencies to identify the impacts and adopt appropriate measures to minimize the negative effects of rail disruptions.

The robustness (resilience, redundancy, vulnerability, etc.) of the URT service partly depends on the topological structure of the URT network, so some studies focused on the effects on the URT system under random disruptions based on graph theory or complex network theory [1,2,3,4,5,6]. The identical nodes or links were identified by analyzing the topological characteristics, such as node degree, betweenness, centrality, etc. For example, Yang et al. [1] assessed the robustness of the Beijing URT network in face of unplanned disruptions. A new node importance evaluation index based on the weighted sum of betweenness and node degree was presented to measure node significance. Jian and Guan [3] proposed a methodology to identify the most influential lines and measure the URT network vulnerability from a line operation perspective with consideration of the disruption probability. Chen et al. [6] developed a demand-impedance indicator and an effective path betweenness index to evaluate the importance of rail stations.

The previous studies belong to the static impact analysis of the URT service disruptions. The dynamic passenger flow was not taken into consideration. In practice, once a disruption occurs, affected passengers would make travel adjustments, including mode shift, resetting original and terminal points, and travel route selection. The travel adjustment choice of the affected passengers at the micro level is finally reflected in the redistribution of passenger flow on the transportation network at the macro level. Thus, some studies were dedicated to understanding the mechanism of passengers’ behaviors and constructing passenger reassignment models under unplanned rail disruptions [7,8,9,10,11,12,13,14,15]. Generally, these works were conducted by means of a passenger flow survey. For instance, Teng and Liu [7] made a passenger behavior survey and a stated preference survey in Shanghai. They found that passengers’ primary choice in response to the interruption is to make a detour in the URT network and that the temporary shuttle bus routes and existing bus lines are also welcomed by passengers as alternative modes. Then a multinomial logit model was built to assign the passenger flow under section interruption in the URT system. An online survey of rail users was conducted by Currie and Muir [8] to understand passenger behaviors, perceptions and priorities during unplanned rail disruptions. An interesting observation was obtained that over two-thirds of passengers choose the replacement buses as their substitute transit mode even though it takes a long time to wait for the bus service to operate. According to Zhu et al. [9], income played a vital role in determining travel reactions to disruptions. Nguyen–Phuoc et al. [10] conducted semi-structured interviews with 30 commuters from Melbourne, Australia, and found the most important factors influencing the mode shift when public transport ceases in the short term, e.g., car access, travel time, travel cost, trip importance, which were also observed by Adelé et al. [11] and Li et al. [12].

On the basis of survey results, logit models were commonly employed to make travel choice behavior analyses. Teng and Liu [7] built a multinomial logit model to assign the passenger flow under section interruption in the URT system. Considering uncertain disruption duration, Li et al. [14] developed a nested logit model to explore metro passengers’ travel plan choice behavior under unplanned service disruptions. Wang et al. [15] established a nested logit model following random regret minimization principles to estimate passenger travel choice behaviors under metro emergency context. Dai et al. [16] also proposed a nested logit model for the decision-making of affected passengers in a metro emergency evacuation.

Apart from survey-based approaches, some scholars used data-driven methods to explore the influence of unplanned rail disruptions on the URT passenger flows. The historical passenger flow data collected from the automated fare collection (AFC) system under normal states and disrupted states were compared and analyzed, hence capturing the passenger volume changes and estimating the impacts of rail disruptions [17,18,19,20,21,22]. For instance, Silva et al. [17] used network-wide data obtained from smart cards in the London transport system to predict future traffic volumes and to estimate the effects of disruptions due to unplanned closures of stations or lines. Sun et al. [18] introduced a Bayesian model to identify disruptions based on automated fare collection (AFC) data, and the effects of disruptions were modeled concerning the affected passengers and the length of delay in the URT network. Through analyzing mobile phone records before and after a collision accident of a metro line in Shanghai, Duan et al. [19] studied the passenger evacuation process and the impacts of the accident on commuters. Based on smart card data, Eltved et al. [20] proposed a method to analyze the effects of long-term disruptions on passenger travel behavior by distinguishing travel behavior changes prior to the disruption and after the disruption.

Deterministic methods or aggregated methods cannot reflect the heterogeneity of passengers’ behaviors under unplanned rail disruptions, while simulation is a better approach to model the heterogeneous passengers. On one hand, agent-based simulation was widely used [23,24,25,26,27,28,29]. Leng and Corman [24] applied agent-based micro-simulation (MATSim) to imitate large-scale passengers’ behaviors during public transport disruptions. A within-day replanning method was integrated into MATSim to study passengers’ route choices. The research revealed the importance of the issue time of information, which greatly impacts passenger satisfaction with public transport disruptions. A similar study was conducted by Muller et al. [26], where MATSim was used to simulate agents’ choice to find an alternative route after a shutdown of an underground line. Based on simulation results with different policies, the effect of the disruption was analyzed, including travel time, number of line changes, and line usage. Li et al. [27] developed a discrete- event simulation model with parallel computing to estimate the effects of metro emergencies. Zargayounaa et al. [28,29] presented a multi agent simulation model to evaluate the impact of real-time information provision on passenger travel times. Results showed that the impact was positive until a threshold was reached.

On the other hand, Monte Carlo (MC) is another simulation method to analyze the uncertainty of passengers’ behaviors. Seger and Kisgyörgy [30] utilized MC simulation to quantify uncertainty in traffic assignment on a transport network. Similarly, the MC technique was applied by Bhat [31] to evaluate the variations in travel mode choice in a multinomial logit model. To examine the day-to-day evolution of network traffic flow, Wang and Li [32] constructed a mixed logit model and designed an MC simulation algorithm to solve the model. Huang et al. [33] proposed an MC-based method to evaluate travel time reliability under the impact of traveler information.

However, there are some limitations in the previous studies. First, the existing works failed to provide a method to accurately identify and estimate the affected passenger flow. Second, concerning passengers’ modal choice behavior, the existing studies only considered passengers’ individual choices based on the utility of each alternative mode, such as cost, travel time, and transfer times. Actually, due to the limited capacity of transit modes, the choices of passengers who access the URT network in advance would affect those of passengers who enter the network later. However, the complex interactions among different passengers and the transit system were neglected in previous research. Third, most previous studies, model-based, or simulation-based, only performed the impact analysis of unplanned rail disruptions on affected rail passengers and rail networks, e.g., passenger volume, delay, travel time, and rail usage. As a matter of fact, the impacts are not limited to the interior of the rail system but extend to the whole transportation system. For instance, buses are a relatively popular choice for affected rail passengers as an alternative transportation mode to complete their travel [8,34]. When a large number of rail passengers shift to the bus system, the original bus passengers are certain to be affected. To the best of our knowledge, no other study focused on the impacts of unplanned rail disruptions on bus passengers.

To fill the research gap, this paper proposes a passenger behavior simulation framework under urban rail disruptions based on the Monte Carlo method and logit model. Affected rail passengers along with conventional bus passengers are modeled as agents that interact with each other and with the transport system. The heterogeneity of passengers’ travel choices can be simulated in a multi-modal public transit network, including the urban rail, bus, taxi, shared bikes and walking. On the basis of simulation, the impacts of the disruptions on rail passengers as well as bus passengers can be analyzed. The results can enable transit agencies to evaluate the passenger evacuation capacity of the present public transit system under unplanned rail disruptions. They can further assist transit management departments in taking appropriate measures to minimize the negative effects on commuters.

The rest of the paper is organized as follows. Section 2 proposes a method to identify the affected passengers under unplanned rail disruptions. Section 3 builds a multi-agent simulation framework to model the travel adjustment behavior of the affected passengers based on the logit model and Monte Carlo method. Then the impact evaluation indices of disruptions are introduced in Section 4. Section 5 discusses a case study. Finally, the conclusions and future research are summarized in Section 6.

## 2. Identification of Affected Rail Passengers

This study focuses on unplanned URT disruptions that result in the closure of rail links between two adjacent turnover stations in both directions [35]. Note that the interrupted section may contain transfer stations or not. The former case, i.e., containing transfer stations, is obviously a more complicated situation, which is studied in-depth in this paper. As shown in Figure 1, Line 1 and Line 2 are both bidirectional rail lines. An unplanned disruption causes service breakdown on a segment of Line 1 including the transfer station A, while Line 2 still functions normally.

The passengers affected by the unplanned disruption refer to those whose space-time travel path in the URT network coincides with the occurrence place and duration of the disruption simultaneously. Under the scenario mentioned before, the affected passengers can be divided into two types in terms of whether they need to transfer in the interrupted line section. In this section, we analyze the time domain of the two types of affected passengers entering the URT system, respectively. On this basis, the affected passenger volume can be estimated using historical automatic fare collection (AFC) data. To identify the affected passengers by their travel route, it is assumed that passengers follow the shortest path to complete their trip.

### 2.1. Notations

S: the set of rail stations, $S=\left\{S_{1},S_{2},\ldots ,S_{n}\right\}$.

$S_{o}$: the origin station of a passenger.

$S_{d}$: the destination station of a passenger.

$S_{h}$: the transfer station in the interrupted section.

$S_{w}$: the two turnover stations adjacent to the interrupted section.

$S_{z}$: the first interrupted station in the forward direction of a passenger’s path.

$S_{l}$: the last interrupted station in the forward direction of a passenger’s path.

$S_{k}$: the transfer station outside the interrupted section and nearest to the interrupted section in a passenger’s detour paths.

$T_{interrupt}$: the time of the occurrence of the rail disruption.

T: the passenger tap-in time.

$t_{interrupt}$: the time duration of the rail disruption.

$T_{recover}$: the time of the recovery of the rail disruption, i.e., $T_{recover}=T_{interrupt}+t_{interrupt}$.

$t_{up}$: the time spent by a passenger from the origin to $S_{z}$ via URT.

$t_{down}$: the time spent by a passenger from $S_{l}$ to the destination via URT.

$t_{spread}$: the time duration from $T_{interrupt}$ to the time at which the disruption information is obtained by passengers.

$t_{i,j}^{metro}$: the time spent by a passenger from $S_{i}$ to $S_{j}$ via URT.

### 2.2. Tap-In Time Domain of Affected Passengers

#### 2.2.1. Affected Non-Transfer Passengers

Affected passengers who need not transfer in the interrupted section can be classified into four categories in terms of the spatial relationship between passengers’ origins and destinations and the interrupted section (see Table 1). Based on the passenger tap-in time, we can identify whether a certain passenger is influenced by the disruption or not. With this regard, the affected tap-in time domain is determined for each category as follows.

Affected passengers of the first category are those whose origins and destinations are both in the interrupted section (i.e., $t_{up}=0$ and $t_{down}=0$). For passengers who enter the URT network before $T_{interrupt}$, the affected time domain is $T_{interrupt}-t_{o,d}^{metro}<T<T_{interrupt}$; for those who enter the network later than $T_{interrupt}$ and depart from the network earlier than $T_{recover}$, the affected time domain is $T_{interrupt}\leq T\leq T_{recover}-t_{o,d}^{metro}$; and for those who enter the network after $T_{interrupt}$ and leave the network after $T_{recover}$, the affected time domain is $T_{interrupt}-t_{o,d}^{metro}<T<T_{recover}$. Consequently, the affected tap-in time domain of the first category is $T_{interrupt}-t_{o,d}^{metro}<T<T_{recover}$. Similarly, the affected time domain of the other three categories can be analyzed. The results are shown in Table 1.

#### 2.2.2. Affected Transfer Passengers

As analyzed to non-transfer passengers, affected passengers who need to transfer in the interrupted section can also be segmented into four categories. Note that the first category is inexistent for affected transfer passengers. The other three categories are explained in Table 2.

### 2.3. Identification Method

#### 2.3.1. Data Preparation

(1)The URT network diagram

We define the URT network as a directed graph G=[V,E] where $V=\{v_{1},v_{2},\ldots v_{n}\}$ is the rail node set, corresponding to rail stations, and $E=\{e_{1},e_{2},\ldots e_{n}\}$ is the rail arc set, representing links between adjacent stations.

(2)The URT disruption data

The data on the rail disruption including the interrupted section and some time-related parameters, i.e., $T_{interrupt}$, $t_{interrupt}$, $T_{recover}$, are assumed to be known.

(3)The shortest paths of all the OD in the URT network

The Dijkstra algorithm is applied to acquire the shortest paths of all the OD in the URT network, which constitute the database of *PathData*. Correspondingly, the travel time of each shortest path is stored into the database. Based on the shortest path of each OD, we can determine its affected category and time domain that are also laid in *PathData.*

(4)The AFC data

After data cleaning, the AFC data of a certain day are stored in the database of *OriginalURTData*. Each piece of data represents a passenger trip, including the passenger ID, the tap-in station ID, the tap-in time, the tap-out station ID, and the tap-out time.

#### 2.3.2. Identification Process

Based on the AFC data and the affected tap-in time domains, affected passengers can be identified. The basic idea is that for a piece of data, we clarify its affected category and compare its tap-in time with the corresponding affected time domain. However, it is tedious and time-consuming to make the comparison for every piece of AFC data, so we simplify the identification process in such a way that the data are sorted in the light of the tap-in time and then some data are excluded using the upper and lower limits of the time domains. The detailed process is listed as follows:

Step 1: Compare the affected tap-in time domains of all OD pairs in *PathData*. Find out the smallest lower limit $T_{\min}^{in}$ and the largest upper limit $T_{\max}^{in}$.

Step 2: Sort the data in *OriginalURTData* according to the tap-in time. Store the data with the tap-in time within $T_{\min}^{in}$ and $T_{\max}^{in}$ into *URTData*.

Step 3: Divide the data in *URTData* into several groups based on the OD. As a result, data with the same OD are included in the same group. Then sort the data in each group according to the tap-in time.

Step 4: For data in each group, find the affected time domain in *PathData*. Then identify the data within the domain and store them into *AffectedURTData*.

## 3. Passenger Behavior under Unplanned Rail Disruption

### 3.1. Trip Plan Selection

Under unplanned rail service disruptions, affected passengers are forced to make adjustments to their trip plans. Figure 2 summarizes the possible passenger behaviors in response to the disruptions. There are five options: (1) changing to an alternative route in the remaining URT network, (2) changing to other modes, (3) using rail and other modes with the latter functions as a “bridge” to restore connectivity between disrupted stations, (4) waiting until the URT system is restored, and (5) delaying or canceling the trip. Other alternative modes include walking, shared bike, bus, and taxi. Note that the response of altering the origin or destination is beyond our consideration. Here Option (4) is valid for passengers who have already entered the URT system before $T_{interrupt}$, while Option (5) is valid to passengers whose T is later than $T_{interrupt}$.

The options for affected passengers of different categories are somewhat different from each other. As shown in Table 3, Options (2), (4) and (5) are available to every passenger category; Options (3) is not optional to only the first category, while Option (1) is available to only the fourth category.

For affected passengers of the fourth category, it should be noted that if passengers have already passed $S_{k}$ when the disruption happens, Option (1) is no longer available to them. Considering that it takes a while for the disruption information dissemination if passengers obtain the information before they reach $S_{k}$, they can resume their travel by detouring in the degraded URT network; otherwise, they miss the detour opportunities due to lack of information. The tap-in time domains in which Option (1) is available to passengers of the fourth category are displayed in Table 4.

In order to determine passengers’ decisions on trip plan selection, the following rules are established:(1)Affected passengers would select Option (1) if feasible detour routes exist in the degraded URT network. This assumption has been proven to be true in reality by Duan et al. [19]. Feasible detour paths shall meet the following two conditions:
the travel time of the path should not exceed a predetermined limit $t_{detour}^{\mathrm{limit}}$.
(1)$$t_{detour}^{metro}\leq t_{detour}^{\mathrm{limit}}.$$the total transfer times of the path should be no more than a predetermined limit $F_{transfer}$.
(2)$$f_{transfer}\leq F_{transfer}.$$(2)Transit agencies can take passenger flow guidance measures to direct passengers with short travel distances to choose slow traffic. With this in mind, we assume that if there is just one station between a passenger’s current located station and his destination station, he will use a shared bike when there are available shared bikes. Additionally, if a passenger’s current located station and his destination station are adjacent stations, passengers will choose walking.(3)The proportion of travelers canceling or delaying their trips is supposed as a given constant.(4)The duration of service disruption is predictable, and it is announced to passengers. Passengers prefer to wait until the system is restored, i.e., select Option (4), if the announced recovery time is less than a predetermined value (denoted as $t_{wait-recover}$.)

### 3.2. Trip Chain Selection

#### 3.2.1. Candidate Trip Chains

Based on the above rules, affected passengers choosing Options (1), (4) and (5) are definite on some specific conditions. However, Options (2) and (3) are uncertain options that heterogeneous passengers may have distinct choices. Comparatively, Option (3) is more complicated in that for different categories of affected passengers, candidate trip chains of Option (3) exhibit differences. Herein, the trip chain means that travel comprises several trip legs, and the transit modes for adjacent legs are different. Especially, Option (2) can be regarded as a trip chain with only one leg.

We further define the starting and ending stations of the trip leg in Options (2) and (3) passed via modes other than rail as $S_{o-new}$ and $S_{d-new}$. Passengers’ decisions on $S_{o-new}$ and $S_{d-new}$ may vary greatly depending on the spatial relationship of their origins and destinations with the interrupted section and their locations at the time of service disruption. According to the locations, affected passengers can be divided into two groups: (i) passengers in and (ii) out of the interrupted rail section.

When $S_{o}$ is in the interrupted section, the position of $S_{o-new}$ can be one of the following three cases: for passengers of Group (i), $S_{o-new}$ will be the dwelling station ($S_{o-new}=S_{current}$) when the train stops at a station under disruption, or the next station ($S_{o-new}=S_{next}$) when the train stops in a section under disruption; for passengers of Group (ii), $S_{o-new}$ will remain to be $S_{o}$ ($S_{o-new}=S_{o}$). When $S_{o}$ is outside the interrupted section, there are also three cases of $S_{o-new}$: for passengers of Group (i), $S_{o-new}$ will also be $S_{current}$ or $S_{next}$ as analyzed before; for passengers of Group (ii), they can take a train to a functional station and then transfer to other modes to pass through the interrupted section, hence resulting in several alternative stations of $S_{o-new}$. In this study, only the most general case in which $S_{o-new}=S_{w}$ is considered.

When $S_{d}$ is in the interrupted section, $S_{d-new}$ remains unchanged as $S_{d}$. When $S_{d}$ is outside the interrupted section, there are many alternatives of $S_{d-new}$ and only the case of $S_{d-new}=S_{w}$ is taken into account here. To sum up, the candidate trip chains of Options (2) and (3) can be derived (see Figure 3). Trip chain ③ belongs to Option (2), while the other five trip chains belong to Option (3). In Figure 3, “pre” and “after” represent the trip leg before and after disruption, respectively. Furthermore, Table 5 summarizes the candidate trip chains for affected passengers of different categories. 

#### 3.2.2. The Logit Model and Monte Carlo Method

Note that “other” in the trip chains means the transit modes other than rail include walking, shared bicycle, bus and taxi, so with distinct transit modes, a trip chain can evolve into various trip chains. Thus, passengers of each category in either Group (i) or (ii) have more than one choice. Here a classical logit model is constructed to describe the trip chain selection for affected passengers [36] (see Equation (3)).
(3)$$P_{od}^{r}=\frac{\exp(-U_{od}^{r})}{\sum_{r\in R}\exp(-U_{od}^{r})}.$$

Given an OD pair, $U_{od}^{r}$ represents the utility of the *r*th trip chain. As seen in Equation (4), it consists of the generalized cost of the *r*th trip chain and a random item. The generalized cost is composed of the money cost and the cost of travel time, which can be divided into in-vehicle time, transfer time and waiting time (see Equation (5)).
(4)$$U_{od}^{r}=u_{od}^{r}+\varepsilon_{od}^{r}.$$
(5)$$u_{od}^{r}=\delta \cdot (t_{od}^{r-invehicle}+\alpha \cdot t_{od}^{r-transfer}+\beta \cdot t_{od}^{r-wait})+c_{od}^{r}.$$
where α denotes the transfer penalty coefficient; β denotes the waiting penalty coefficient; and δ denotes the value of time.

Deterministic methods ignore the randomness in passengers’ travel adjustment behavior, which may result in estimation bias [22]. In this study, the Monte Carlo method is utilized to address the issue of uncertainty in passengers’ trip chain selection. Specifically, for each passenger who has to select a trip chain, we perform a sampling based on the logit model, i.e., generate a random number representing the index of a trip chain according to the proportions of trip chains [37,38,39]. In this study, the Monte Carlo method can be taken as a subroutine of the multi-agent simulation introduced in the next section.

### 3.3. Multi-Agent Simulation Model

Passengers’ trip chain selection depends on the availability of the transport modes. Specifically, passengers can only choose transport modes with residual capacity. Note that the residual capacity at a certain time is influenced by passengers’ selection behavior before that time. In other words, passengers’ decision-making on trip chain selection interacts with each other. To better reflect this interaction, a multi-agent simulation model is developed to simulate the process of passengers’ travel adjustment behavior in a multimodal transport system with dynamic changes.

#### 3.3.1. Multi-Agent System

Considering the complexity and dynamics of the multi-modal transport system, it is preferable to split the system into several independent and interrelated parts [40]. Each part is regarded as a type of agent, including the passenger agent, vehicle agent and station agent. Information is able to be exchanged among these agents. The logical framework of the multi-agent simulation system is shown in Figure 4. The basic idea is that first, the database is constructed, then the simulation experiment is carried out, and finally, the impacts of an unplanned rail disruption are estimated and output.

The basic data, including the infrastructure data, passenger data, and vehicle data, are predetermined as input information for the simulation system. The infrastructure data mainly comprises information on the disrupted rail stations and related bus stations. The passenger data include attributes of affected rail passengers and original bus passengers, e.g., the origin station, the destination station and the tap-in time. The vehicle in the simulation system consists of trains, buses, taxis and shared bikes. The train and bus data mainly involve the capacity, the load factor, the remaining capacity and the timetable, while the taxi and shared bike data mainly contain the number of available vehicles, the turnover rate and the located station.

#### 3.3.2. Passenger Agent

As the fundamental simulation unit, the passenger agent connects and interacts with other agents in the multi-agent simulation model. Passenger agents are classified into two types in this paper: rail passenger agents and bus passenger agents, representing the affected rail passengers and the original bus passengers in the real world, respectively.

For the rail passenger agents, the simulation model mainly simulates the process of train matching, the process of trip chain selection, and the process of using alternative modes. The logical flow chart of the rail passenger agent simulation is displayed in Figure 5. As analyzed before, passengers selecting Options (1), (4) and (5) can be determined by rules, so they are first picked up from the affected passengers. In order to reduce the simulation complexity, this study mainly focuses on the simulation of passengers in the interrupted section. Thus, passengers not entering the interrupted section, e.g., those choosing trip chains ③ and ④ in Group (ii) are also excluded beforehand. Then other passengers are put into the main simulation program. Train matching determines $S_{o-new}$ and the time passengers arrive at $S_{o-new}$ based on the train timetable and passenger tap-in time. Trip chain selection is realized by the logit model and Monte Carlo method stated in Section 3.2.2. Additionally, the passenger agents choosing the bus as their substitute mode will be integrated with bus agents and proceed with the waiting, boarding and alighting process.

Bus passenger agents have the function of waiting for bus, boarding the bus and alighting the bus (see Figure 6). The bus passenger agent is matched with a bus agent according to the time passengers reach the station, the bus timetable, the bus residual capacity and its sequence in the waiting queue. The waiting time of each bus passenger agent can be determined and recorded for the subsequent impact analysis.

#### 3.3.3. Vehicle Agent

As the carrier for the passenger agent, the vehicle agent is to realize the function of simulating the boarding and alighting process by interacting with the passenger agent and the infrastructure agent. Vehicle agents can be divided into train agents, bus agents, taxi agents and share bike agents. 

The train agent mainly realizes the departure function and arrival function based on the train timetable. When a train arrives at a station, it will dwell for a while. Subsequently, rail passenger alighting and boarding events will be triggered. In addition, when the scheduled departure time is up, the departure function is called. As depicted in Figure 7, the bus agent also performs arrival and departure functions. It is worth noting that both bus passenger agents and rail passenger agents who choose the bus as the substitute mode to pass through the interrupted section are involved in the bus operation simulation. Unlike train agents and bus agents, one taxi agent or shared bike agent represents several taxis and bikes at a certain station in the real world, so both train agents and bus agents have the attribute of residual capacity indicating the number of available carriers.

## 4. Impact Estimation of Unplanned Rail Disruptions

In evaluating the impacts of unplanned rail disruptions, our main concern is the impacts on passengers, including rail passengers and non-rail passengers. Note that the existing parallel bus system is one of the first alternatives for affected rail passengers to resume their trips. The original bus travelers are certain to be influenced when a large number of rail passengers shift to the bus system. Thus, for the impacts on non-rail passengers, we focus on the delay time of bus travelers caused by rail disruptions. As indicated before, our simulation focuses on passengers in the interrupted section, so here only these passengers are taken into the following impact estimation.

### 4.1. Estimation of the Impacts on Rail Passengers

(1)Affected rail passenger flow

As noted before, affected rail passengers refer to their travel paths intersecting with the rail disrupted section in space and the rail disruption occurrence period in time. The affected rail passenger flow, denoted as $N_{metro}^{affacted}$, is the total number of affected rail passengers. This index can be estimated using the approach proposed in Section 2.

(2)The stranded time

Herein, the stranded time of the *i*th affected rail passenger, denoted as $t_{stranded}^{i}$, is defined as the time duration the passenger is detained outside a rail station until he is taken away by a certain transit mode. In particular, for passengers who choose walking or shared bike to resume their trips, their stranded time is set to be 0. The total number of affected rail passengers in the interrupted section is assumed to be $N_{metro}^{affacted-in}$. Consequently, the total stranded time can be formulated as follows:(6)$$t_{stranded-in}^{total}=\sum_{i=1}^{N_{metro}^{affacted-in}}t_{stranded}^{i}.$$

The average and maximum stranded time can also be formulated as Equations (7) and (8), respectively.
(7)tstranded−in¯=tstranded−intotalNmetroaffacted−in.
(8)$$t_{stranded-in}^{\max}=\max\left\{t_{stranded}^{i}\right\}.$$

Compared to the delay time commonly defined as the increase in travel time under disruptions, the stranded time indicating the additional waiting time for alternative transit services is more concerned by affected rail passengers. On the other hand, transport authorities are also more interested in the stranded time which suggests the evacuating speed of affected passengers than the travel time increase [41].

(3)Stranded rail passenger flow

Due to the limited capacity of available transport services, some affected rail passengers may be stranded outside a rail station until the disrupted rail service is recovered. Let $N_{metro}^{stranded-in}$ be the number of these stranded passengers in the interrupted section and the corresponding proportion can be calculated as follows:(9)$$\alpha_{metro}^{stranded-in}=\frac{N_{metro}^{stranded-in}}{N_{metro}^{affacted-in}}.$$

The larger $\alpha_{metro}^{stranded-in}$ is, the greater the gap between the residual capacity of the available transit modes and the scale of affected rail passengers. It suggests that other substitute transport services are required to evacuate stranded passengers, such as the bus bridging service.

### 4.2. Estimation of the Impacts on Original Bus Passengers

(1)The delay time

With the increased ridership caused by the modal shift of affected rail passengers, bus passengers would be delayed at the bus stops, resulting in a longer wait time for the original bus passengers waiting for an available bus. $t_{wait-bus}^{i,j}$ representing the waiting time of the *i*th bus passenger at the *j*th bus stop in the interrupted section is commonly calculated as half of the headway of the bus line that he desires to take. Suppose $t_{wait-bus}^{i,j}{}^{\prime }$ be the actual waiting time under the rail disruption. Then the delay time of the *i*th bus passenger can be formulated as the increase in the waiting time (see Equation (10)).
(10)$$\Delta t_{wait-bus}^{i,j}=t_{wait-bus}^{i,j}{}^{\prime }-t_{wait-bus}^{i,j}.$$

The total delay time can be estimated by the following equation:(11)$$\Delta t_{wait-bus}^{total}=\sum_{j=1}^{N_{stop}}\sum_{i=1}^{V_{j}}\Delta t_{wait-bus}^{i,j}.$$
where $N_{stop}$ is the number of bus stops and $V_{j}$ is the number of bus passengers at the *j*th bus stop during the rail disruption.

The average and maximum delay time can be obtained by Equations (12) and (13).
(12)$$\bar{\Delta t_{wait-bus}}=\Delta t_{wait-bus}^{total}/\sum_{j=1}^{N_{stop}}V_{j}.$$
(13)$$\Delta t_{wait-bus}^{\max}=\max\left\{\Delta t_{wait-bus}^{i,j}\right\}.$$

(2)Affected bus passenger flow

Suppose bus headway to be $h_{bus}$. If $t_{wait-bus}^{i,j}{}^{\prime }-h_{bus}>0$, it reveals that the *i*th bus passenger is affected by the rail disruption. On this basis, a binary variable $\eta_{i,j}$ is introduced to indicate whether the *i*th bus passenger at the *j*th bus stop in the interrupted section is affected (see Equation (14)).
(14)$$\eta_{i,j}=\left\{\begin{array}{ll} 1, & if\;t_{wait-bus}^{i,j}{}^{\prime }-h_{bus}>0;\\ 0, & otherwise. \end{array}\right..$$

Subsequently, the total number of bus passengers in the interrupted section who are affected by the disruption can be estimated by Equation (15).
(15)$$N_{bus}^{affacted-in}=\sum_{j=1}^{N_{stop}}\sum_{i=1}^{V_{j}}\eta_{i,j}.$$

(3)Stranded bus passenger flow

Since affected rail passengers take up some bus resources, several affected bus passengers would be unable to board buses and are stranded at bus stops until the disrupted rail service is recovered. Let $\omega_{i,j}$ be a binary variable indicating whether the *i*th bus passenger at the *j*th bus stop in the interrupted section is finally stranded. If yes, $\omega_{i,j}=1$; otherwise, $\omega_{i,j}=0$.
(16)$$N_{bus}^{stranded-in}=\sum_{j=1}^{N_{stop}}\sum_{i=1}^{V_{j}}\omega_{i,j}.$$

## 5. Case Study

In this section, part of the Ningbo URT network is employed to conduct a case study. As shown in Figure 8, the test network consists of 3 bi-directional rail lines and 75 stations, of which 3 are transfer stations.

### 5.1. Scenario Description

Two scenarios of unplanned rail disruption are designed here to make a comparison. As seen in Table 6, both of the interrupted sections in the two scenarios contain the transfer station $S_{10}$. Compared with Scenario B, Scenario A is a comparatively smaller disruption with fewer interrupted stations. Additionally, the disruption in Scenario A happens in non-peak hours of a working day, while that in Scenario B arises in peak hours. The historical AFC data of a typical weekday are stored in the database of *OriginalURTData* and applied to make the impact estimation.

As indicated before, this study focuses on the simulation of passengers in the interrupted section. Therefore, three bus lines parallel to the interrupted section are integrated into the simulation system, and the impact analysis on the original bus passengers also targets these three bus lines. Related data on these bus routes, including data on passengers as well as buses, are stored in the database of the simulation system. Referring to the existing research [8,22,42], some parameters in the simulation system are assigned as listed in Table 7.

### 5.2. Results Analysis

Since the simulation involves a stochastic trip chain selection process by the Monte Carlo method, the results of each simulation are distinct. We perform the simulation 50 times. It appears that there are small differences among those simulation results. For instance, regarding the average stranded time for rail passengers, the simulation results of Scenario B are between 14.97 and 15.53, with a standard deviation of 0.15. Thus, the average value of 50 simulation results is taken as the estimation result of each indicator. Some proportion-related data are given using a certain simulation result.

#### 5.2.1. Impact Estimation Results for Rail Passengers

(1)Estimation of affected rail passenger flow

Using the identification method described in Section 2.3, we obtain the estimated results of $N_{metro}^{affacted}$ for scenarios A and B as 6959 and 22,734, respectively. Figure 9 depicts more specific information on the affected rail passenger flow composition, i.e., the number of passengers and OD pairs in the four categories. It can be concluded that passengers of the fourth category take the majority of the total affected rail passengers. Therefore, emergency buses should be dispatched not only to the interrupted rail stations but also to stations of other uninterrupted rail lines. This finding is in line with the observations of Duan et al. [17].

The results of trip plan selection and trip chain selection for affected rail passengers are shown in Figure 10 and Table 8, respectively. The result of Scenario A is illustrated using the inner circle in Figure 10, while that of Scenario B is illustrated using the outer circle. It can be seen that the outcomes of the two scenarios have small differences. Options (1) and (3) take the majority in both scenarios, indicating that most affected rail passengers would continue to use the degraded URT network to resume their travel [17].

To further analyze passengers in the interrupted section, the outbound passengers at each interrupted station needing to be evacuated by a certain transportation mode other than rail are described in Figure 11. As can be seen, most affected rail passengers depart from the URT system at the transfer station in the interrupted section as well as the turnover stations. This suggests that emergency buses should be dispatched to these stations to evacuate passengers and avoid secondary accidents caused by passenger gathering. Furthermore, Figure 11 shows the modal split results of the “other” mode in the trip chains. Additionally, a special type of passengers who are stranded until the disrupted rail service is recovered is also added into Figure 11. Based on the preset rules listed in Section 3.1, there are many passengers walking to their destination stations. It relieves the pressure of passenger evacuation on other transit modes, e.g., buses, taxis. This confirms that passenger flow guidance is extremely important in response to unplanned disruptions.

(2)Estimation of the stranded time and stranded rail passenger flow

As illustrated in Table 9, in Scenario B, more than 35% of affected rail passengers do not be taken away by other transit modes until the disrupted rail service resumes normal operations. This results in an excessively long waiting time for passengers detained at interrupted stations. As can be seen, the maximum stranded time reaches 58.97 min.

Furthermore, the distribution of stranded time is depicted in Figure 12. Passengers with stranded time exceeding 10 min account for approximately 3% in Scenario A, while the proportion in Scenario B is about 47%. Additionally, even more than 3000 passengers detained for over 30 min. Obviously, the alternative transit modes are not able to evacuate affected rail passengers efficiently and effectively in Scenario B, indicating that other emergency buses are in urgent demand.

#### 5.2.2. Impact Estimation Results for Original Bus Passengers

As anticipated, with a large number of rail passengers shifting to the bus system, the original bus passengers are more or less influenced. As seen in Table 10, bus passengers in Scenario B are affected to a greater extent than those in Scenario A in terms of the estimation results of delay time and the affected bus passenger flow. Specifically, the average time passengers wait to board buses increases by about 11 min in Scenario B. The affected bus passengers number over 1000, taking up approximately 57% of all boarding passengers in the interrupted section. Moreover, there are still 577 bus passengers who are unable to board buses and are stranded at bus stops until the disrupted rail service is recovered. In contrast, there are no bus passengers detained in Scenario A. Figure 13 presents the delay time distribution of the original bus passengers. The results further confirm the fact that bus passengers in Scenario B are greatly affected. The bus passengers with a delay time of more than 10 min account for about 34%. Additionally, there are over 100 bus passengers delayed over 30 min.

To further investigate the impacts on bus passengers at different bus stops, Table 11 and Table 12 are plotted illustrating the passenger waiting time with passenger arrival time at stops for Scenarios A and B, respectively. In Scenario A, the waiting time of most passengers is within the bus headway except for passengers at the transfer station S_10_. Comparatively, for passengers at more stops, including the transfer station, turnover stations and intermediate stations, wait times for buses exceeds bus headway in Scenario B, especially for passengers at stops in the upstream direction. These most affected bus stops need to be paid more attention by transit agencies. Note that in these bus stops, the passenger waiting time follows the rule that it gradually increases with the time of arrival and then displays a linear decrease after it reaches the maximum value. It is because the continuous accumulation of arriving passengers results in incremental waiting time for passengers. Passengers whose arrival time is prior to the inflection point can board buses before $T_{recover}$. Their wait time gradually increases. Passengers whose arrival time is after the inflection point are finally detained at stops until the disrupted rail service is recovered. Their waiting time gradually declines based on the assumption that the affected rail passengers would return to the rail system after $T_{recover}$ and the detained bus passengers are able to board the first arriving buses. It also warns us that with regard to the bus stop with a large number of stranded passengers, the passengers reaching the stop after $T_{recover}$ may still be affected.

## 6. Conclusions

Evaluating the impacts of unplanned urban rail disruptions on the transportation system enables transit agencies to determine the severity of the disruptions and adopt effective measures to mitigate the negative effects and ensure passengers’ safety. Previous studies paid much attention to evaluating the impacts on the rail system itself, but little attention to the potential effects on the entire public transport system. To fill the gap, this study proposes a simulation-based approach to estimate the impacts on public transport travelers, including rail travelers as well as bus travelers. Firstly, we present a method of identifying affected rail passengers and estimating the scale and origin-destination distribution of affected passengers. Then a multi-agent simulation model is developed to simulate the multimodal transport evacuation process. Original bus passengers are integrated into the simulation system and interact with the rail passengers. On the basis, some indicators evaluating the impacts on rail passengers and bus passengers are presented respectively. Finally, a case study based on the Ningbo URT network is conducted and some practical results are acquired.

Whether simulation or impact assessment, this study only focuses on passengers in the interrupted section, since they are influenced by the disruptions most immediately and to the greatest extent. As analyzed in Section 2, the affected rail passengers can spread all over the URT network. Correspondingly, affected bus passengers are also not limited to the interrupted section. If the research scope is expanded to the whole public transport network, much more bus lines and bus passengers would be added to the simulation system. The issue becomes more complicated, and it is our ongoing work. Concurrently, the scale of rail disruptions in the study can be enlarged to an entire line or even a partial network.

## Figures and Tables

**Figure 1 ijerph-19-09052-f001:**
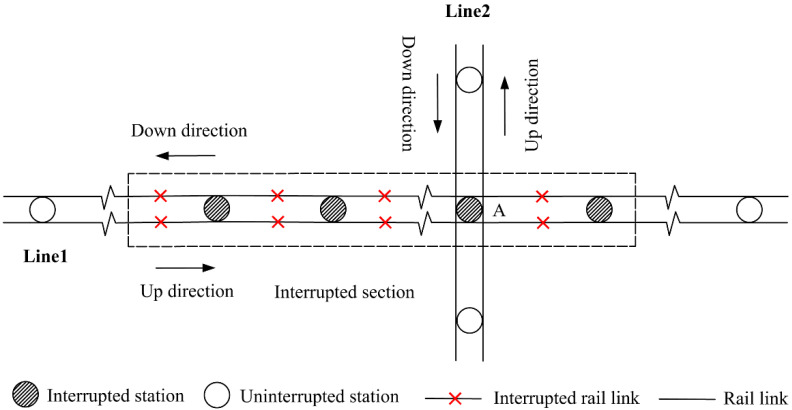
Illustrative example of URT service disruptions.

**Figure 2 ijerph-19-09052-f002:**
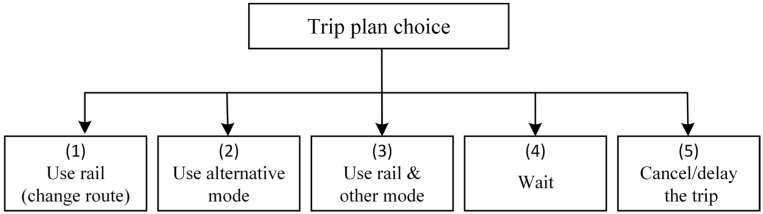
Possible trip plans for affected passengers.

**Figure 3 ijerph-19-09052-f003:**
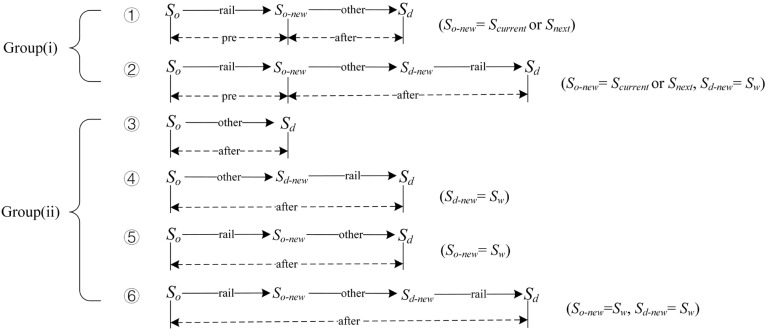
Candidate trip chains.

**Figure 4 ijerph-19-09052-f004:**
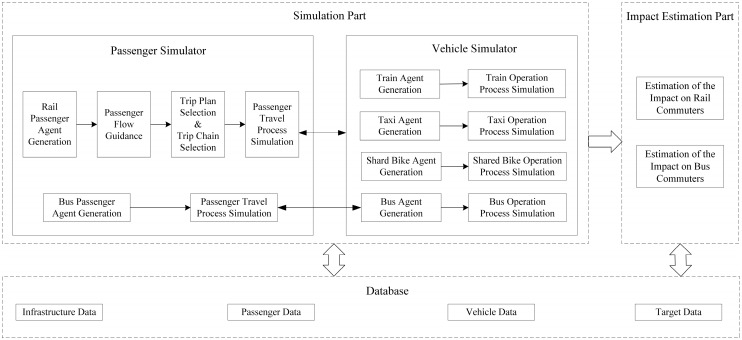
The logical framework of the multi-agent simulation system.

**Figure 5 ijerph-19-09052-f005:**
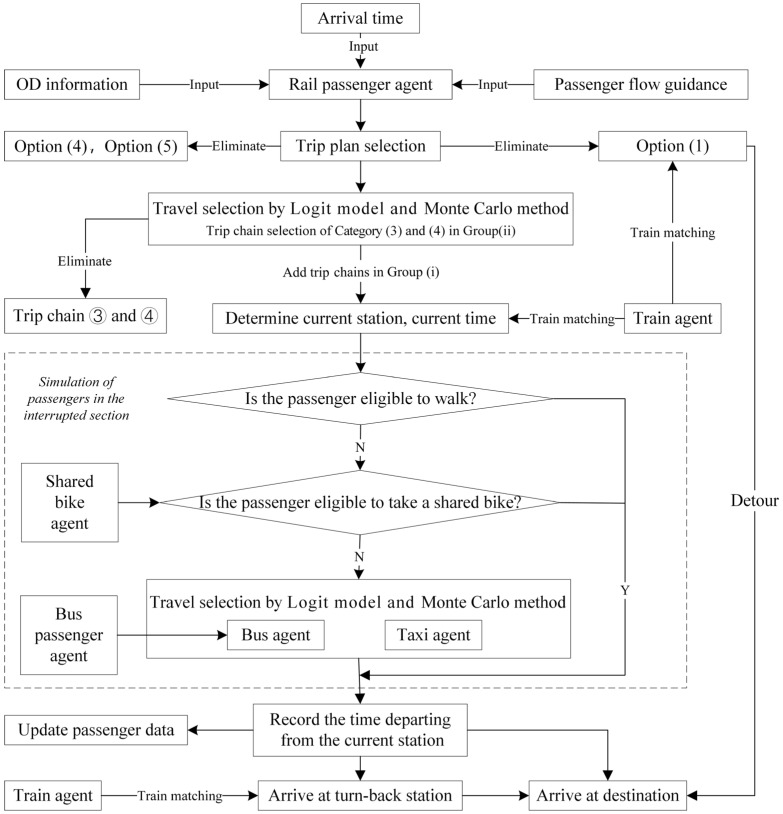
Rail passenger agent main function.

**Figure 6 ijerph-19-09052-f006:**
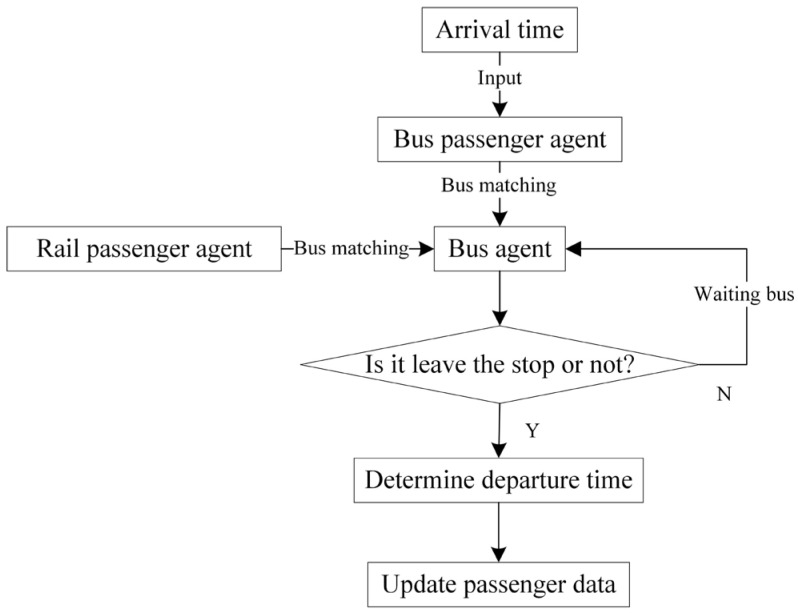
Main function of bus passenger agent.

**Figure 7 ijerph-19-09052-f007:**
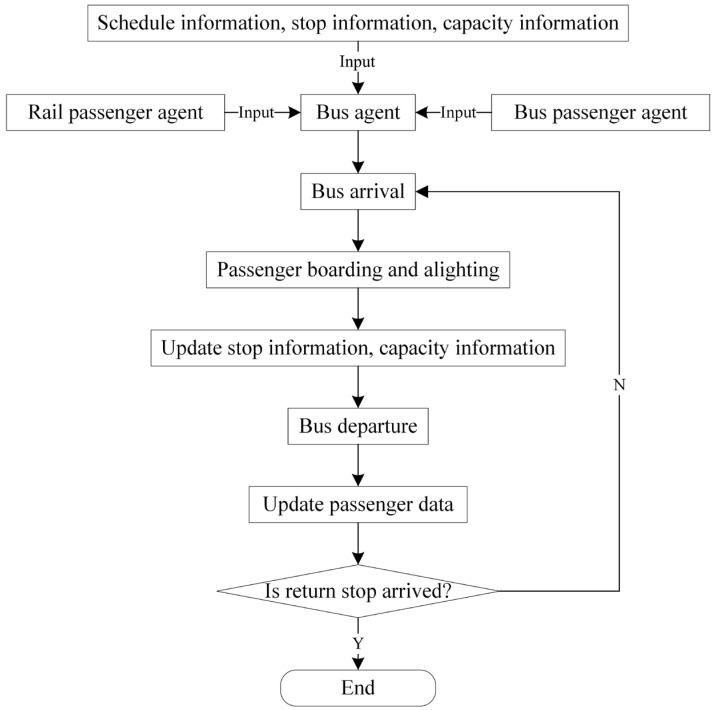
The logical flow chart of bus agent operation process simulation.

**Figure 8 ijerph-19-09052-f008:**
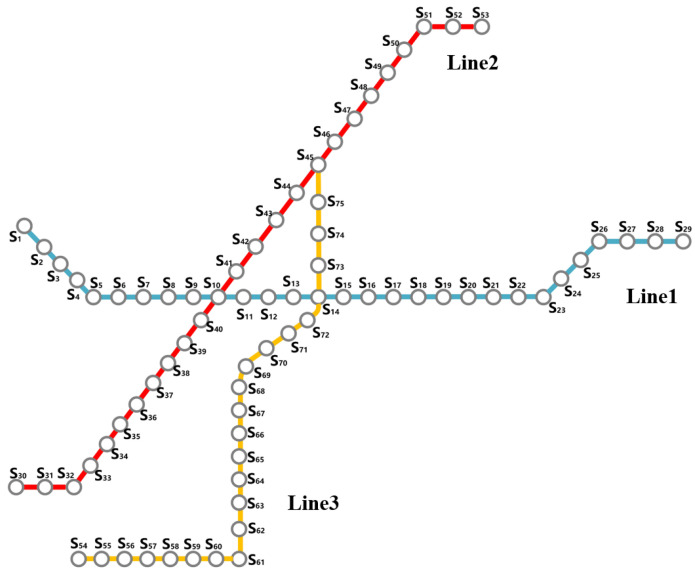
Map of the partial URT network in Ningbo.

**Figure 9 ijerph-19-09052-f009:**
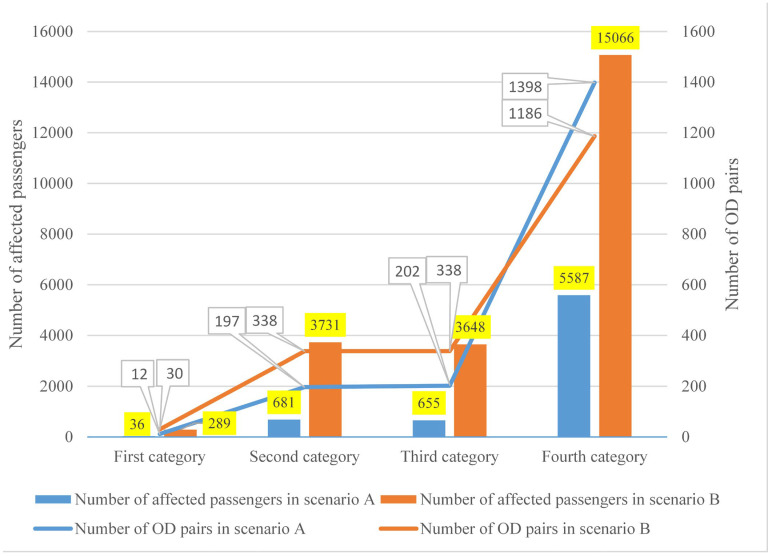
Number of affected rail passengers and OD pairs in different categories.

**Figure 10 ijerph-19-09052-f010:**
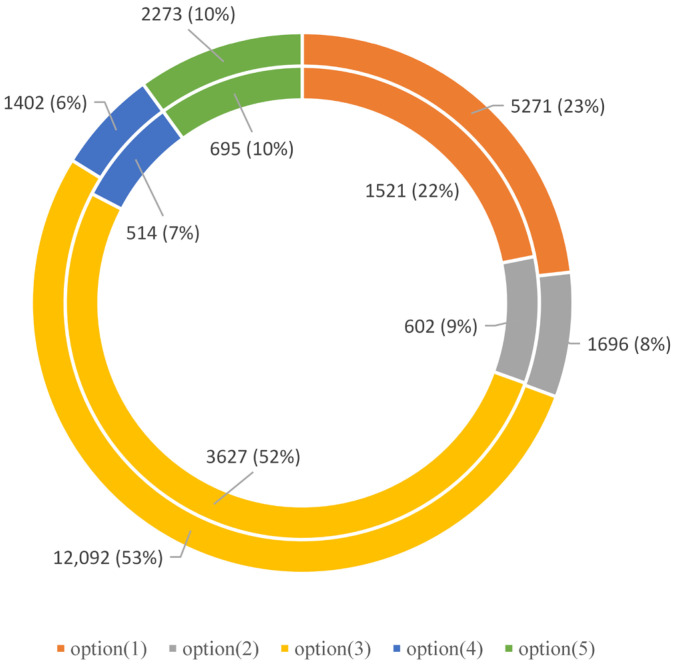
Trip plan selection results.

**Figure 11 ijerph-19-09052-f011:**
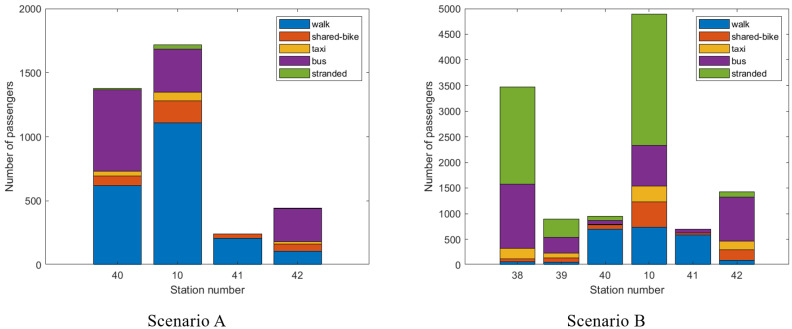
Travel selection results at each interrupted station.

**Figure 12 ijerph-19-09052-f012:**
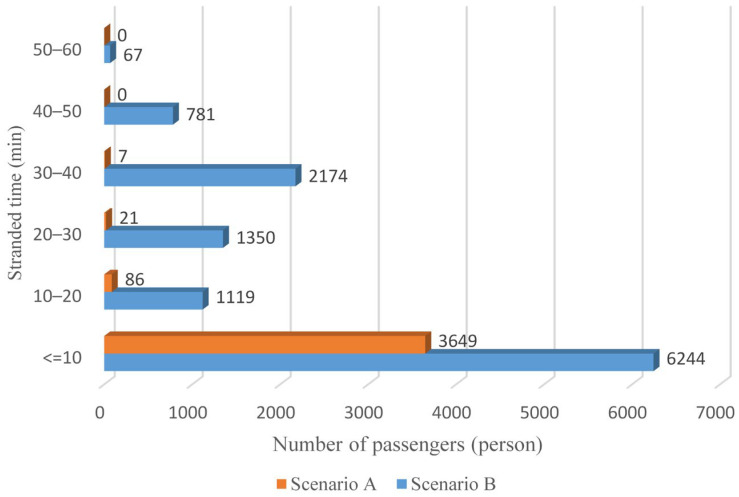
Stranded time distribution of affected rail passengers.

**Figure 13 ijerph-19-09052-f013:**
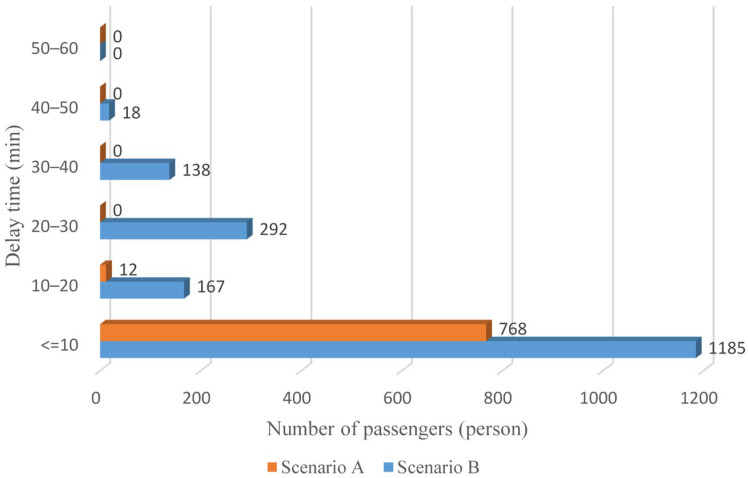
Delay time distribution of original bus passengers.

**Table 1 ijerph-19-09052-t001:** The tap-in time domains of affected non-transfer passengers.

Passenger category	$\mathrm{The}\;\mathrm{first}\;\mathrm{category}:\;t_{up}=0$ $\mathrm{and}\;t_{down}=0$	$\mathrm{The}\;\sec\mathrm{ond}\;\mathrm{category}:\;t_{up}=0$ $\mathrm{and}\;t_{down}>0$
Time domain	$T_{interrupt}-t_{o,d}^{metro}<T<T_{recover}$	$T_{interrupt}-t_{o,w}^{metro}<T<T_{recover}$
Illustration	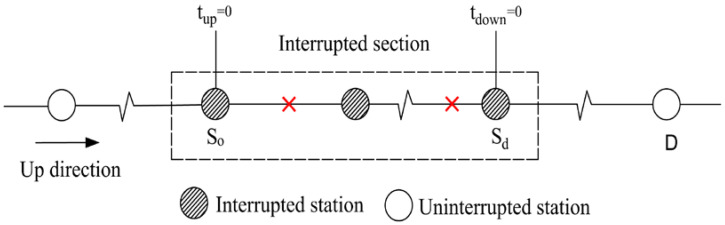	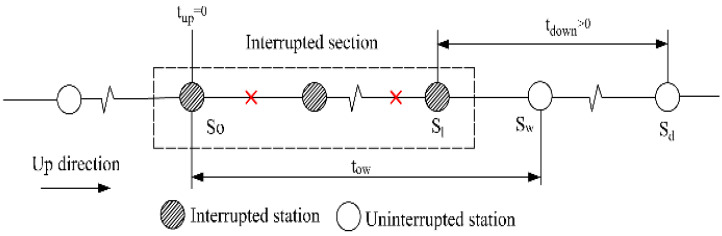
Passenger category	$\mathrm{The}\;\mathrm{third}\;\mathrm{category}:\;t_{up}>0$ $\mathrm{and}\;t_{down}=0$	$\mathrm{The}\;\mathrm{fourth}\;\mathrm{category}:\;t_{up}>0$ $\mathrm{and}\;t_{down}>0$
Time domain	$T_{interrupt}-t_{o,d}^{metro}<T<T_{recover}-t_{o,z}^{metro}$	$T_{interrupt}-t_{o,w}^{metro}<T<T_{recover}-t_{o,z}^{metro}$
Illustration	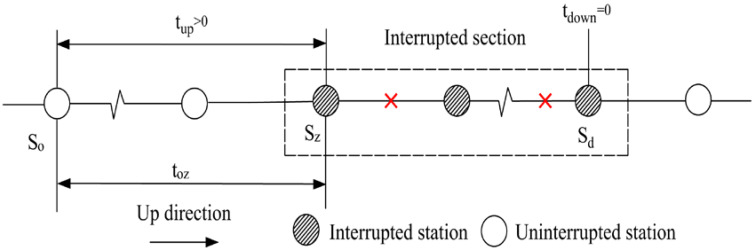	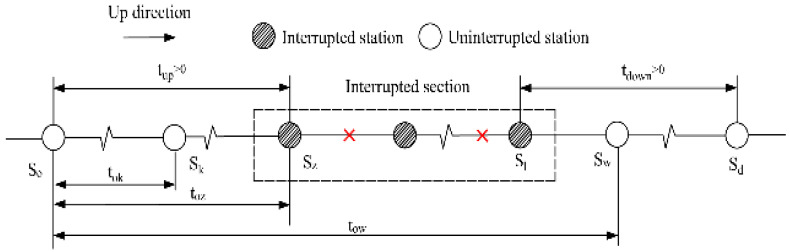

**Table 2 ijerph-19-09052-t002:** The tap-in time domains of affected transfer passengers.

Passenger category	$\mathrm{The}\;\mathrm{second}\;\mathrm{category}:\;t_{up}=0$ $\mathrm{and}\;t_{down}>0$	$\mathrm{The}\;\mathrm{third}\;\mathrm{category}:\;t_{up}>0$ $\mathrm{and}\;t_{down}=0$
Time domain	$T_{interrupt}-t_{o,w}^{metro}<T<T_{recover}$	$T_{interrupt}-t_{o,d}^{metro}<T<T_{recover}-t_{o,z}^{metro}$
Illustration	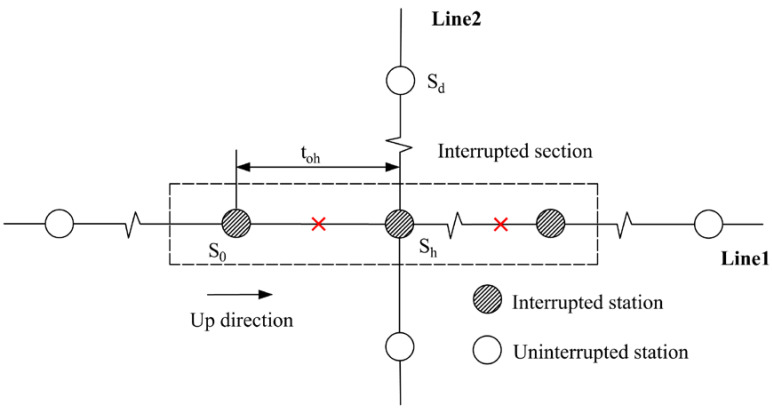	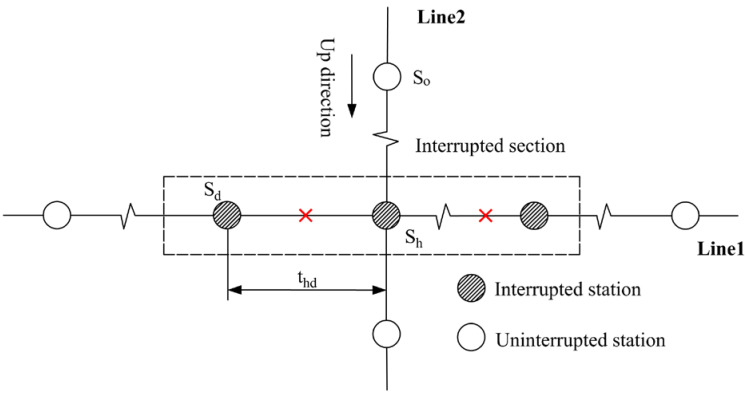
Passenger category	$\mathrm{The}\;\mathrm{fourth}\;\mathrm{category}:\;t_{up}>0$$\mathrm{and}\;t_{down}=0$ (transfer from Line 1 to Line 2)	$\mathrm{The}\;\mathrm{fourth}\;\mathrm{category}:\;t_{up}>0$$\mathrm{and}\;t_{down}>0$(transfer from Line 2 to Line 1)
Time domain	$T_{interrupt}-t_{o,h}^{metro}<T<T_{recover}-t_{o,z}^{metro}$	$T_{interrupt}-t_{o,w}^{metro}<T<T_{recover}-t_{o,h}^{metro}-t_{h}^{walk}$
Illustration	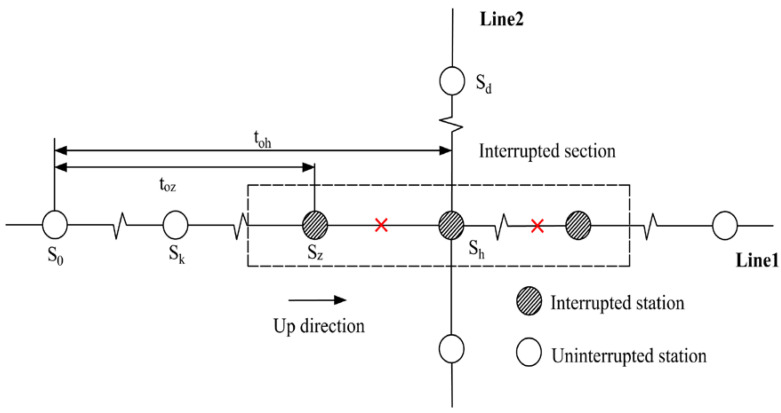	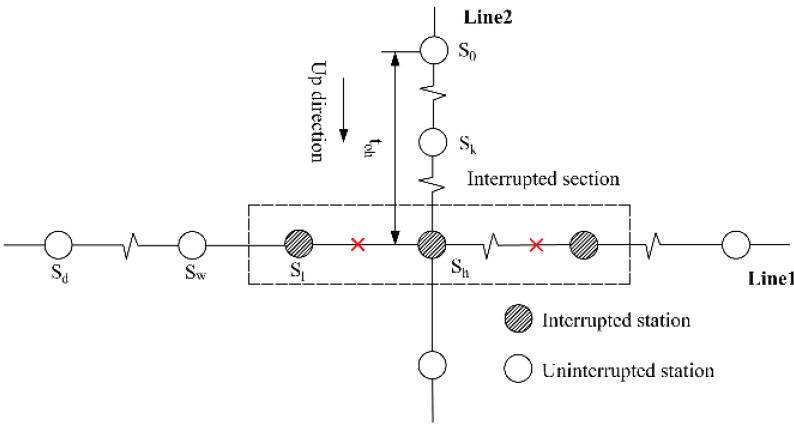

**Table 3 ijerph-19-09052-t003:** The options for affected passengers of different categories.

Passenger Category	Option (1)	Option (2)	Option (3)	Option (4)	Option (5)
The first category		√		√	√
The second category		√	√	√	√
The third category		√	√	√	√
The fourth category	√	√	√	√	√

**Table 4 ijerph-19-09052-t004:** The tap-in time domains in which Option (1) is available to passengers of the fourth category.

Passengers of the Fourth Category		The Tap-In Time Domain
Non-transfer passengers		$T+t_{spread}-t_{o,k}^{metro}\leq T<T_{recover}-t_{o,z}^{metro}$
Transfer passengers	From Line 1 to Line 2	$T+t_{spread}-t_{o,k}^{metro}\leq T<T_{recover}-t_{o,z}^{metro}$
	From Line 2 to Line 1	$T+t_{spread}-t_{o,k}^{metro}\leq T<T_{recover}-t_{o,h}^{metro}-t_{h}^{walk}$

**Table 5 ijerph-19-09052-t005:** Candidate trip chains for affected passengers of different categories.

		Category (1)	Category (2)	Category (3)	Category (4)
Group (i)	Trip chain ①	√	√	√	√
	Trip chain ②		√		√
Group (ii)	Trip chain ③	√	√	√	√
	Trip chain ④		√		√
	Trip chain ⑤			√	√
	Trip chain ⑥				√

**Table 6 ijerph-19-09052-t006:** Basic information of the two scenarios.

	Scenario A	Scenario B
Interrupted section	$S_{40}\Leftrightarrow S_{42}$	$S_{38}\Leftrightarrow S_{42}$
$T_{interrupt}$	11:00 a.m.	8:00 a.m.
$t_{interrupt}$	1 h	1 h

**Table 7 ijerph-19-09052-t007:** Assigned values of some parameters in the simulation.

Parameters	Assigned Values	Parameters	Assigned Values
Bus capacity	80 persons	$t_{spread}$	10 min
$t_{velocity}^{bike}$	10 min	Number of available share bikes at a station	50
$t_{velocity}^{taxi}$	3 min	Number of available taxis at a station	5
$t_{detour}^{\mathrm{limit}}$	$1.5t_{non-detour}^{\mathrm{limit}}$ ^1^	$F_{transfer}$	3
$\gamma_{option(5)}$	10%	$t_{wait-recover}$	10 min
α	2	β	2.5
δ	40 CNY/h		

^1^ $t_{non-detour}^{\mathrm{limit}}$ represents the travel time of shortest path.

**Table 8 ijerph-19-09052-t008:** Trip chain selection results.

	Category (1)		Category (2)		Category (3)		Category (4)	
	A	B	A	B	A	B	A	B
Trip chain ①	3	18	13	43	257	861	299	1028
Trip chain ②			38	103			1498	4025
Trip chain ③	26	93	137	564	65	291	82	394
Trip chain ④			370	1167			319	1368
Trip chain ⑤					161	784	263	874
Trip chain ⑥							698	2175

**Table 9 ijerph-19-09052-t009:** Estimation results of stranded time and stranded rail passenger flow.

Scenario	$t_{stranded-in}^{total}$	$\bar{t_{stranded-in}}$	$t_{stranded-in}^{\max}$	$\alpha_{metro}^{stranded-in}$
A	5652.84 min	1.50 min	39.00 min	1.41%
B	178,992.83 min	15.25 min	58.97 min	35.84%

**Table 10 ijerph-19-09052-t010:** Estimation results of delay time and affected bus passenger flow.

Scenario	$\Delta t_{wait-bus}^{total}$	$\bar{\Delta t_{wait-bus}}$	$\Delta t_{wait-bus}^{\max}$	$N_{bus}^{affacted-in}$	$N_{bus}^{stranded-in}$
A	295.91 min	0.38 min	13.30 min	100	0
B	19,967.60 min	11.09 min	45.44 min	1030	577

**Table 11 ijerph-19-09052-t011:** Distribution of waiting time per bus passenger in Scenario A.

Scenario	Station	Upstream	Station	Downstream
A	40	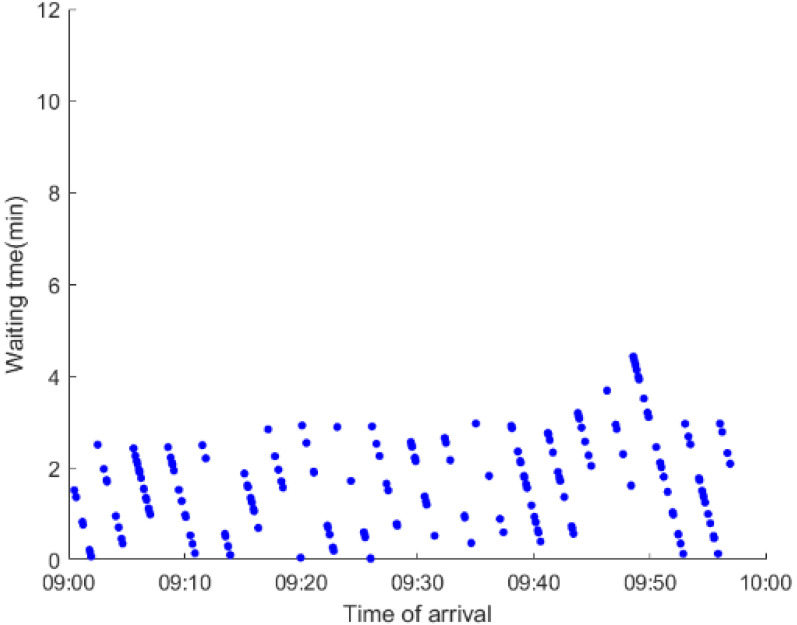	42	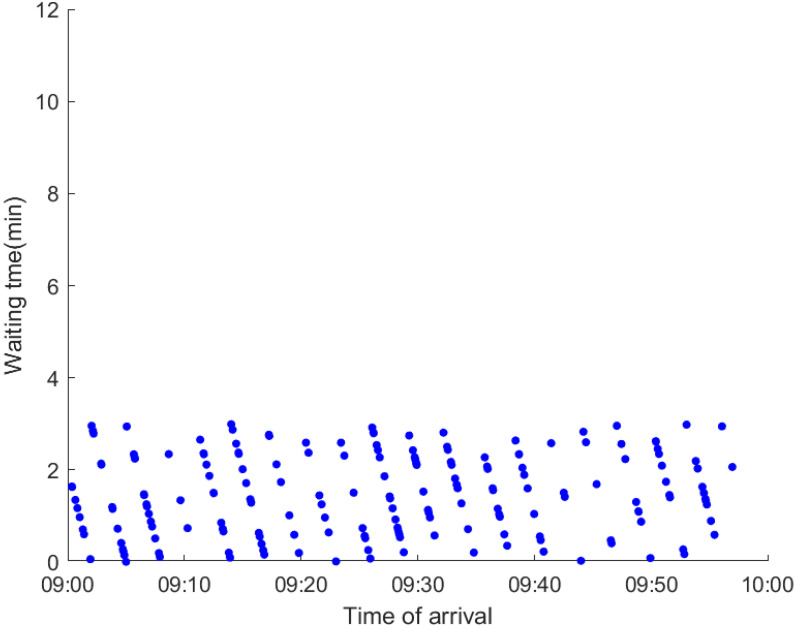
	10	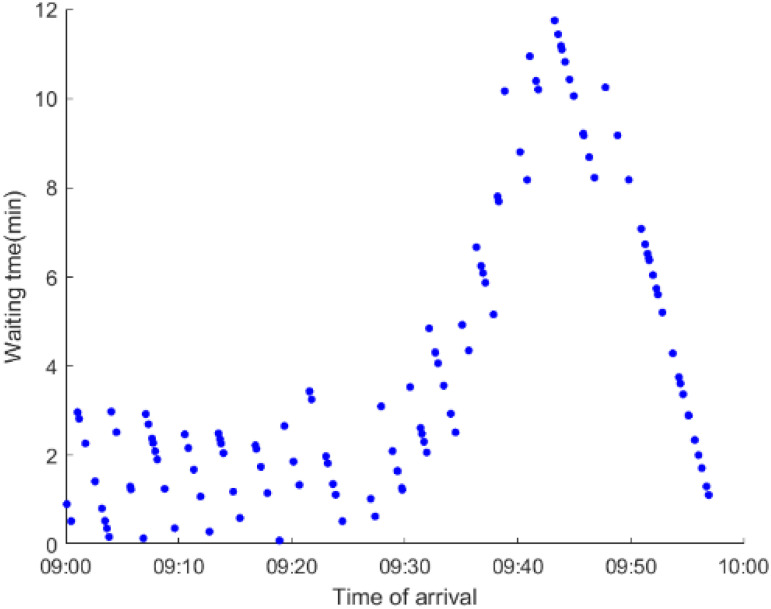	41	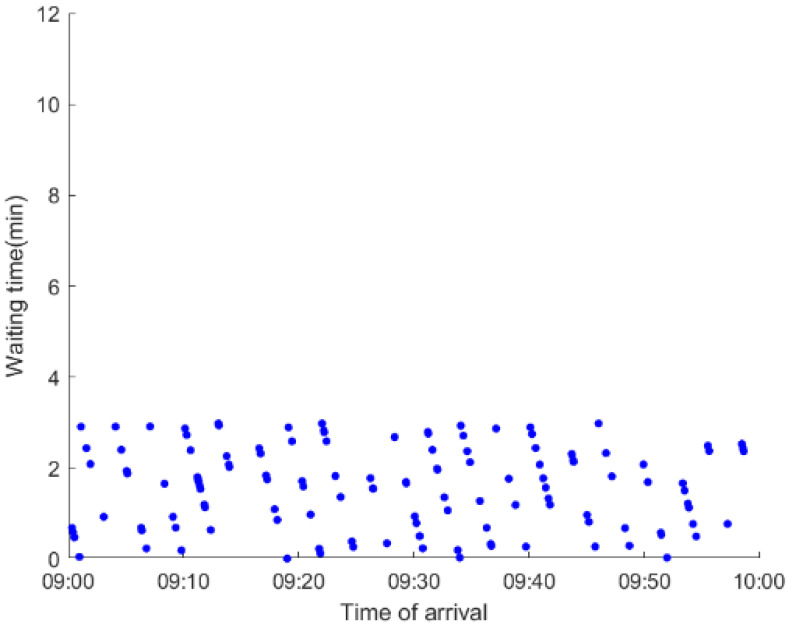
	41	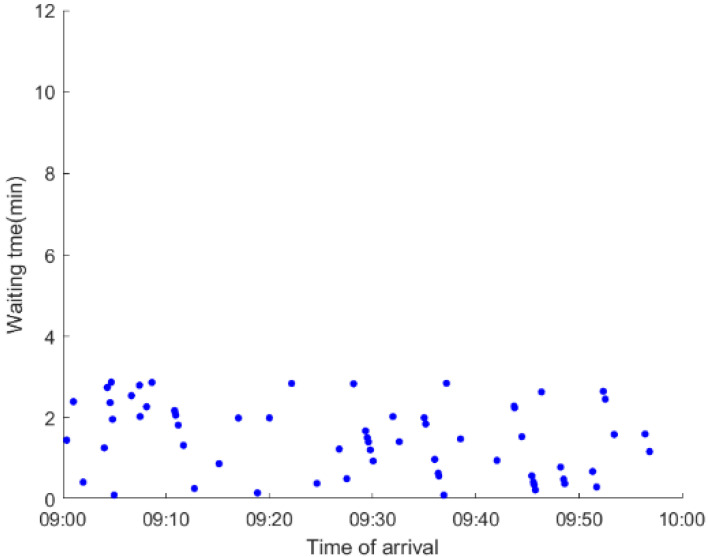	10	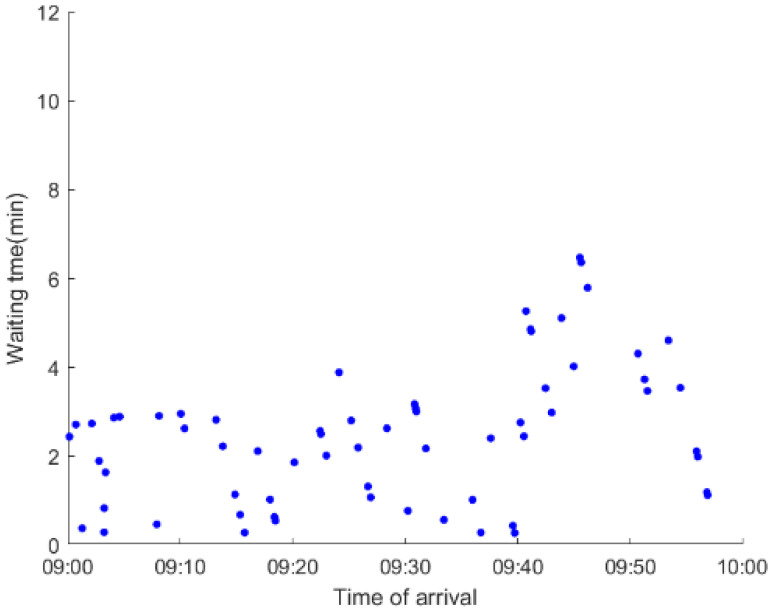

**Table 12 ijerph-19-09052-t012:** Distribution of waiting time per bus passenger in Scenario B.

Scenario	Station	Upstream	Station	Downstream
B	38	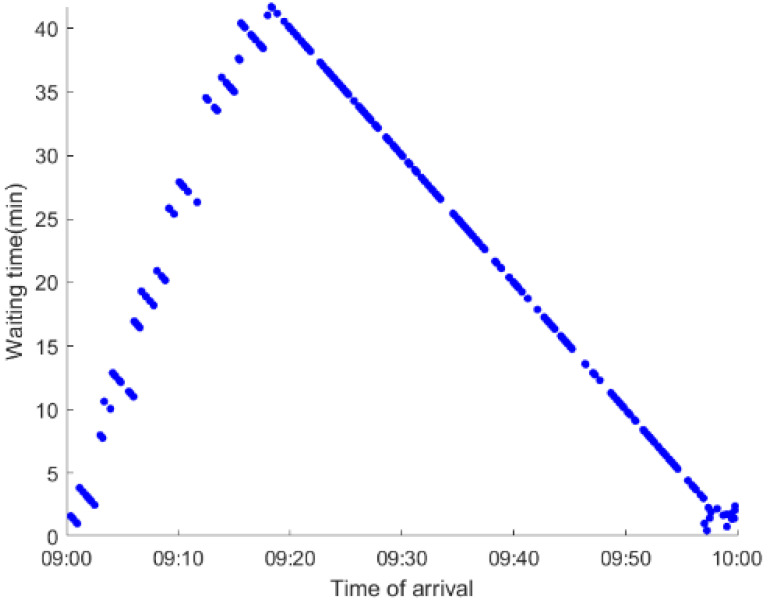	42	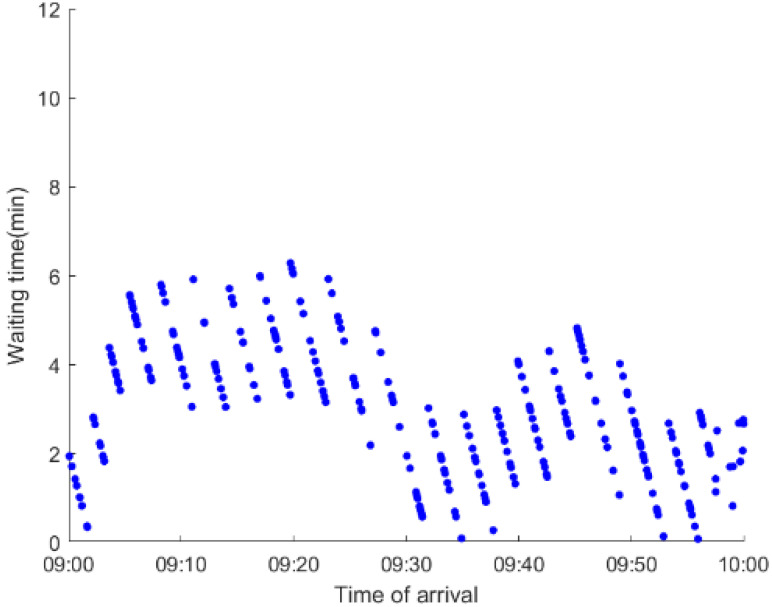
	39	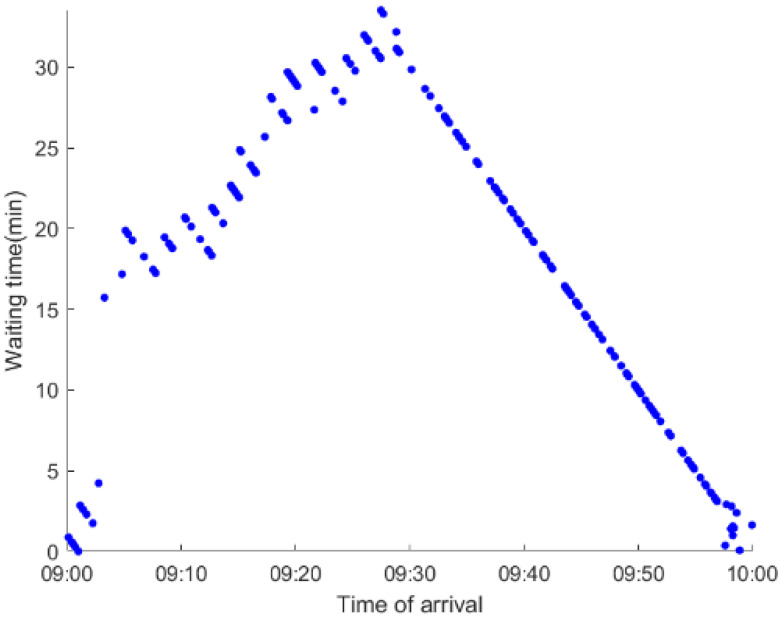	41	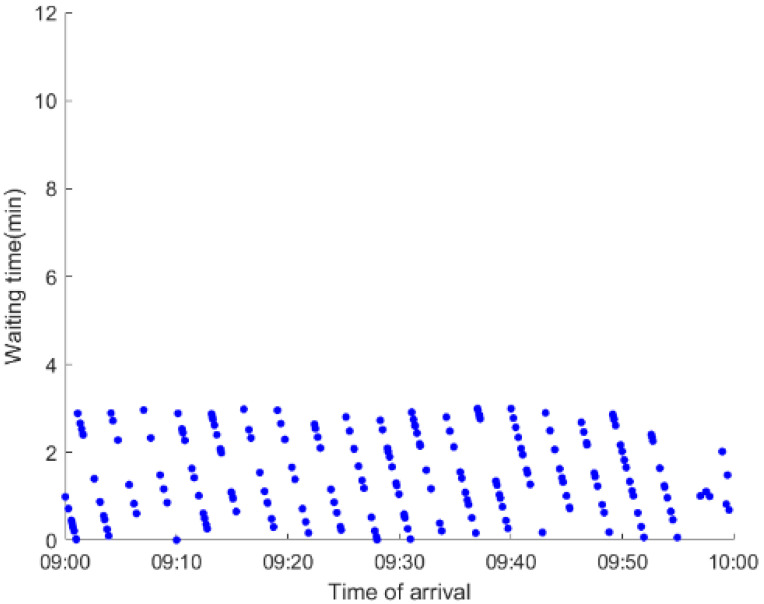
	40	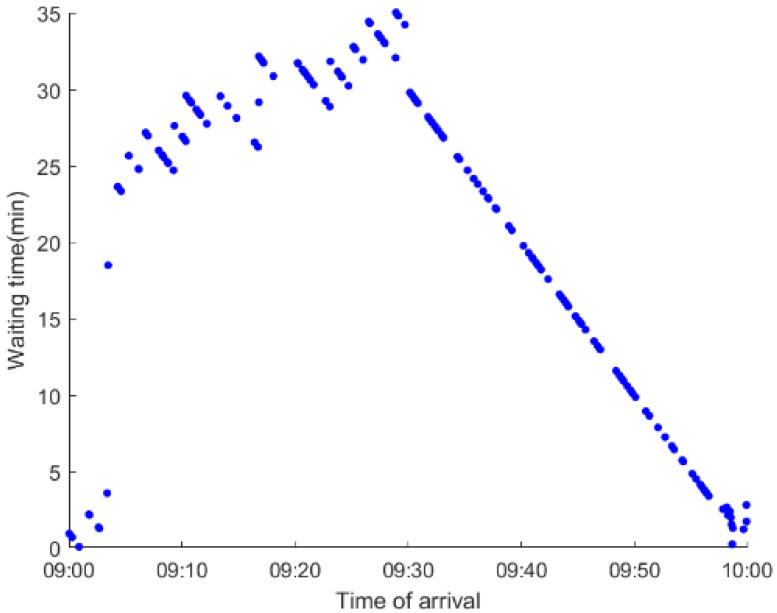	10	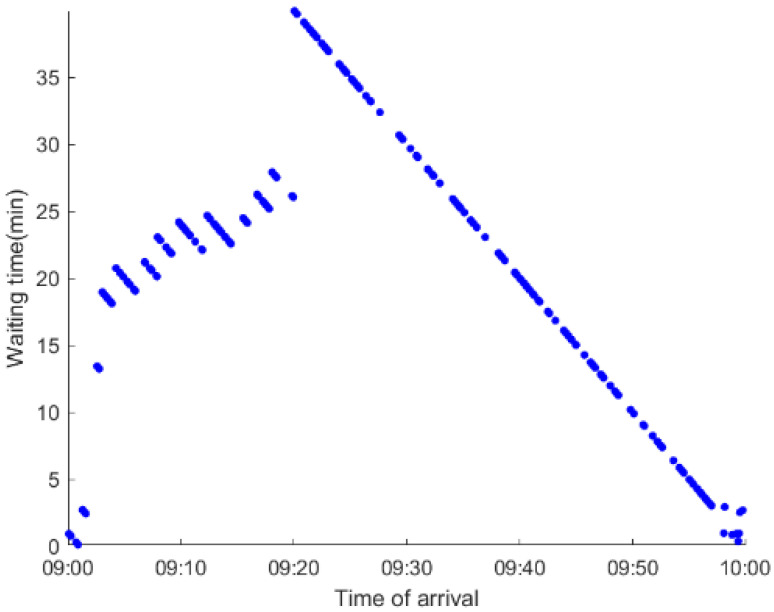
	10	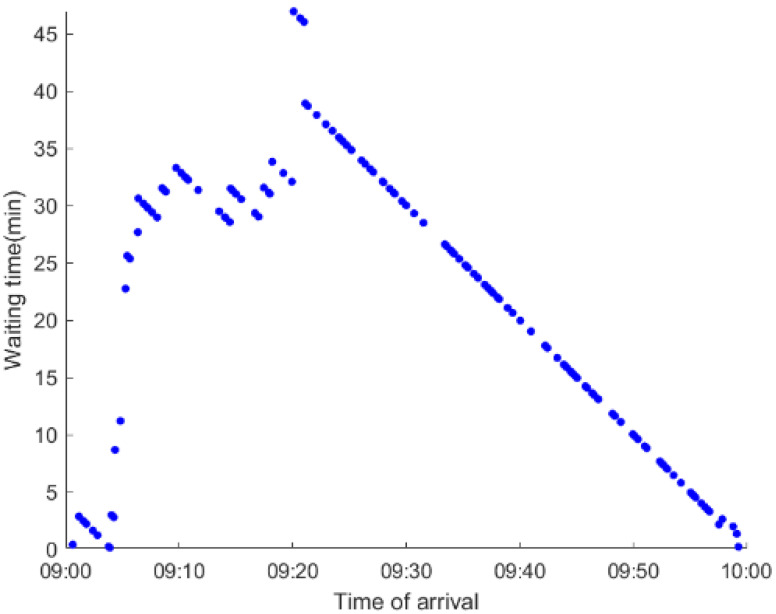	40	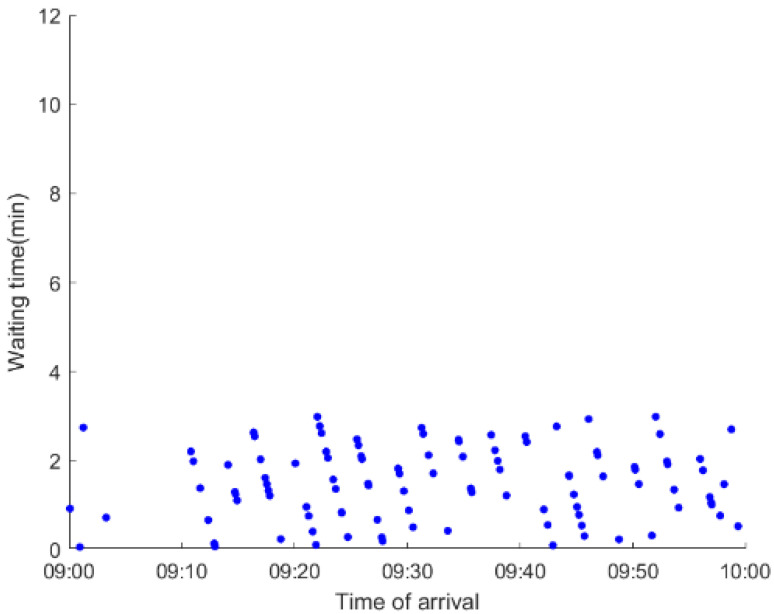
	41	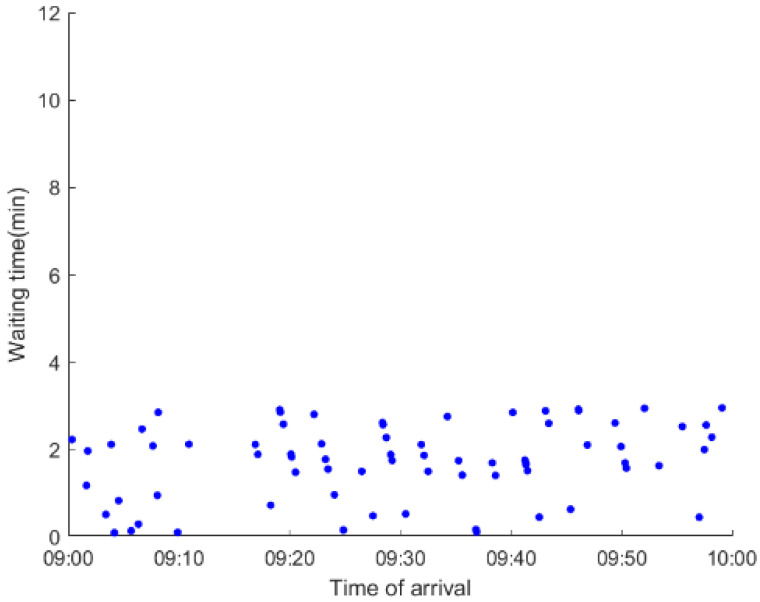	39	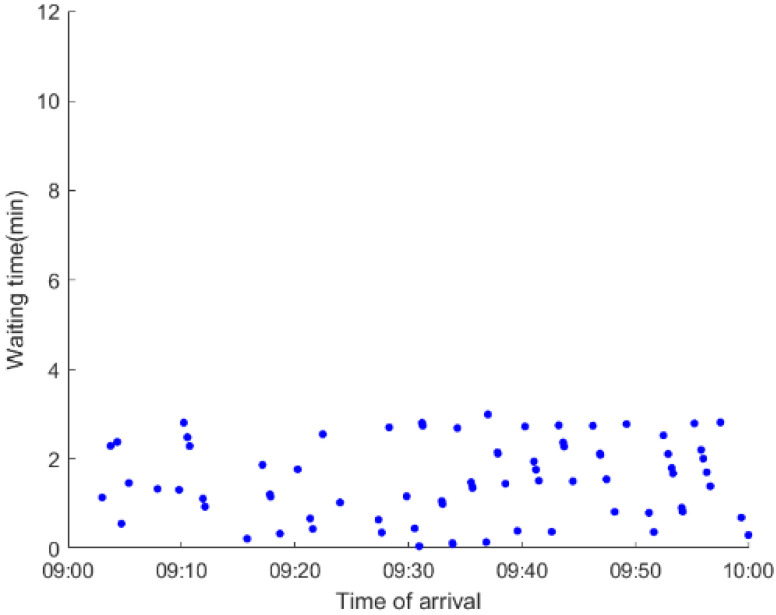

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
