# Peer review of "Impact Estimation of Unplanned Urban Rail Disruptions on Public Transport Passengers: A Multi-Agent Based Simulation Approach"

_ijerph, 2022, doi:10.3390/ijerph19159052_

Round 1

Reviewer 1 Report

This paper attempts to estimate the impacts of unplanned rail line disruptions on rail passengers and original bus passengers. The authors proposed a method of identifying affected rail passengers based on passenger tap-in time and used Ningbo transit network data as a case study. The general subject underlying the study is interesting. However, the manuscript should go through detailed revision to be considered for publication. Some comments are provided below.

The rational for the study is clear. However, the literature review in the current form is more like listing related publications rather than synthesizing them into the introduction. Moreover, the authors should consider including related publications of logit model and Monte Carlo on this topic.

The proposed methodology is introduced well, with clear visualizations and detailed explanation.

Line 264, how is  determined?

Figure 9 isn’t really necessary in this context. It’s known that stochastic simulation will produce distinct result each time, and I think providing the numeric summary is enough (i.e., range, standard deviation). If the authors decide to keep it, figure 9 should be replotted to improve the quality.

Overall, the theme of this manuscript is interesting but writing of this paper is unclear, which poses difficulties for readers to understand. Especially the introduction part and the literature review are unstructured. The language quality should definitely be improved (for example, line 49 “The dynamic passenger flow were not taken into consideration.”; line 65 “conducted a semi-structured interviews with 30 commuters”; use of articles a/an/the throughout the paper). In addition, many awkward word choices should be reconsidered.

Reviewer 2 Report

In this paper, the authors evaluate the effect of disruptions in urban rail networks. They try to assess the potential effect on the entire public transport system. To do so, the study adopts a multi-agent simulation-based approach. They first identify the affected rail passengers. Then they simulate the multimodal transport evacuation process. A case study based is conducted, and some experimental results are presented. 

The paper is generally well written and well structured. The adopted methods are sound, and the results are well described and interpreted. 

However, I see many common points with a non-referenced work, which proposes a multi-agent simulation of disruptions with scenarios on passengers' information level. As far as I understood, they consider both rail and bus passengers. I believe this work should be cited and discussed. 

See "Multiagent simulation of real-time passenger information on transit networks, IEEE ITS Magazine, 2020" and "Impact of travelers information level on disturbed transit networks: a multiagent simulation, IEEE ITSC, 2015"
